# Novel Insights on Pyoverdine: From Biosynthesis to Biotechnological Application

**DOI:** 10.3390/ijms231911507

**Published:** 2022-09-29

**Authors:** Filippo Dell’Anno, Giovanni Andrea Vitale, Carmine Buonocore, Laura Vitale, Fortunato Palma Esposito, Daniela Coppola, Gerardo Della Sala, Pietro Tedesco, Donatella de Pascale

**Affiliations:** Department of Eco-Sustainable Marine Biotechnology, Stazione Zoologica Anton Dohrn, Via Ammiraglio Ferdinando Acton 55, 80133 Naples, Italy

**Keywords:** pyoverdine, siderophore, iron, *Pseudomonas* sp., mass spectrometry

## Abstract

Pyoverdines (PVDs) are a class of siderophores produced mostly by members of the genus *Pseudomonas*. Their primary function is to accumulate, mobilize, and transport iron necessary for cell metabolism. Moreover, PVDs also play a crucial role in microbes’ survival by mediating biofilm formation and virulence. In this review, we reorganize the information produced in recent years regarding PVDs biosynthesis and pathogenic mechanisms, since PVDs are extremely valuable compounds. Additionally, we summarize the therapeutic applications deriving from the PVDs’ use and focus on their role as therapeutic target themselves. We assess the current biotechnological applications of different sectors and evaluate the state-of-the-art technology relating to the use of synthetic biology tools for pathway engineering. Finally, we review the most recent methods and techniques capable of identifying such molecules in complex matrices for drug-discovery purposes.

## 1. Pyoverdines’ General Features

Pyoverdines (PVDs) are fluorescent molecules produced by bacteria belonging to the genus *Pseudomonas* (Figure 1I). Also known during the last decades as fluoresceins and pseudobactins, they were discovered at the end of the nineteenth century [1]. However, the understanding of their biological functions in microbial metabolism remained unknown until 1978, the year in which Meyer and Abdallah [2] and Meyer and Hornsperger [3] unraveled their role in the acquisition of iron. Therefore, from a functional point of view, PVDs belong to the class of siderophores, small molecules with a mass between 200 and 2000 Da that are able to chelate iron and other metals [4].

Iron is a crucial element for the metabolism and survival of microorganisms as it is involved in mechanisms such as the reduction of oxygen for the synthesis of ATP and DNA precursors [5,6]. Despite iron being one of the most widespread elements on our planet, its bioavailability is greatly reduced due to the reduced solubility of Fe (III) oxyhydroxide particulates, the prevailing form at neutral pH and in oxygenated environments [7].

Therefore, PVDs, and more generally siderophores, being very soluble molecules and having a high affinity with iron, exert their main role by taking up the iron present in the extracellular environment to internalize it.

From the literature, five classes of siderophores are known, each of which differs from the functional group used to bind the metals: catecholate, hydroxamate, salicylate, carboxylate, and mixed-type (Table 1).

The three most represented groups of siderophores are catecholates, carboxylates, and hydroxamates [40].

PVDs belong to the mixed-type category, and like many other siderophores, are non-ribosomal peptides. Such molecules originate from non-ribosomal peptide synthetases (NRPSs), large multi-modular enzymes capable of catalysing the synthesis of peptides without an RNA template [41]. The organization in modules of the NRPSs generally contributes to the possibility of producing a wide spectrum of bioactive compounds [42] and most likely influences the specific species diversity of the different PVDs as each one is characterized by a unique peptide moiety [43]. Such a feature enables each *Pseudomonas* strain to acquire iron through the PVDs released by itself, or at least through heterologous PVDs sharing high homology with the native PVD [44,45], since all PVDs, once the iron is bound, only bind to a specific receptor (ferripyoverdine, FpvA) placed on the outer bacterial membrane [46]. The chemical structure of PVDs can be summarized in three parts: (i) the chromophore core (Figure 1II); (ii) the peptide backbone; and (iii) the side chain (Figure 1III). The most conserved structural element in PVDs is the chromophore, a 1S)-5-amino-2,3-dihydro-8,9-dihydroxy-1H-pyrimido-[1,2-a] quinolone-1-carboxylic acid (Figure 1III), with an absorbance at 400 nm (at neutral pH) and an emission of fluorescence at 447 nm.

However, some biological precursors and derivatives display structural modifications, such as isopyoverdine (IsoPy), dihydro-pyoverdine (DiHPy), dihyodro-pyover-dine-7-sulfonics acid (SPy), ferribactin (FerB), Azotobactin (AzoB), and succinopyoverdine (SuccPy) [47].

The other two structural features of PVDs include a variable acyl side chain attached to the 3- amino group of the fluorophore, and a strain-specific peptide backbone, usually bound to the C1 -carboxyl group of the ring system. Interestingly, the strain-specific peptide backbone varies in its sequence and can either be linear or (partially) intramolecularly cyclized [48].

Notably, the PVDs’ peptide chain is fundamental to bind iron. Indeed, PVDs chelate Fe (III) due to three bidentate chelating residues: (i) a catechol group; (ii) a hydroxamate group at the end of the peptide chain; and (iii) an hydroxamate group in the middle of the peptide chain [49,50].

Overall, the PVDs’ great affinity for iron and, more generally, metals aroused great interest among the scientific community due to their promising application in the field of human health and environment care. To provide all the information necessary for a possible biotechnological use, we will describe the main steps of PVD biosynthesis in the following paragraph.

## 2. PVD Biosynthesis

Generally, the fluorescent *Pseudomonas* species produce PVDs as major siderophores to access iron. In particular, *P. aeruginosa* strains produce three different PVDs (PVDI, PVDII, and PVDIII), each one characterized by a different peptide chain [51].

During the past few years, many reviews investigating PVD biosynthesis have been published [43,48,52,53,54,55]. In this paragraph we describe, in a general way, the role of the different enzymes involved in the biosynthesis of PVD type 1 (PVDI) produced by *P. aeruginosa*, focusing on the most recent data of the last two years.

As stated above, PVDs are synthesized by NPRSs organized into modules (from a minimum of 2 to a maximum of 18), each of which selects and catalyzes the addition of a specific substrate to the final product. The mechanism that allows the elongation of the nascent chain is based on the presence, in each module, of three core domains: (i) adenylation (A) domain, which selects a cognate amino acid and converts it to the corresponding aminoacyl adenylate; (ii) peptidyl carrier protein (PCP) domain (also known as thiolation domain), carrying the growing peptidyl chain via a phosphopantetheinyl arm; (iii) condensation (C) domain, which catalyses the peptide bond formation [56]. The complete peptide chain is then released from the multi-enzymatic complex following the action of the thioesterase domain, present only in the termination module [57].

PVD biogenesis starts in the cytoplasm, where its biological precursor, i.e., an acylated ferribactin, is assembled by three multimodular NRPS (PvdL, PvdI, and PvdD) together with several tailoring enzymes. The four-module NRPS PvdL catalyses the biosynthesis of the chromophore precursor. The first module of PvdL recruits myristic or myristiloeic acid as a starter unit, which, as reported by [58], is supposed to prevent the diffusion of the peptide beyond the membrane during assembly. The fatty acid unit is then transferred to the second module of PvdL, catalysing the acylation of an L-glutamate residue. The next two PvdL modules promote the sequential incorporation of D-Tyrosine (D-Tyr) and L-2,4 Diaminobutyrate (L-Dab), thus forming a tetrahydropyrimidine ring, i.e., the precursor of the chromophore [43,59].

Subsequently, the peptide is modified by the action of PvdI, an enzyme composed of four modules catalyzing the addition of D-Ser, L-Arg, D-Ser, and L-N5-formyl-N5-hydroxyornithine (L-hfOrn) [60]. Other auxiliary enzymes associated with the biosynthetic pathway are PvdH, PvdA, and PvdF, responsible for the formation of specific amino acids composing the PVDs backbone, such as L-Dab and L-hfOrn. In detail, PvdH promotes the L-Dab synthesis starting from L-aspartate β-semialdehyde, while PvdA and PvdF allow L-ornithine hydroxylation and formylation yielding L-fOHOrn [61,62]. Interestingly, Gasser and colleagues [63] showed, using the FRET–FLIM (Förster resonance energy transfer measured by fluorescence lifetime microscopy) technique, that PvdA interacts physically with PvdJ, PvdI, PvdL, and PvD, suggesting the presence of a strongly organized multienzymatic complex coordinating the PVD biosynthesis. Similarly, Philem and collaborators [64] demonstrated the fundamental role played by PvdF through a mutagenesis experiment, as the mutants carrying the inactivated enzyme had a reduced production of PVDI. Therefore, the authors suggest a possible switch to regulate PVD production for biotechnological purposes.

As just mentioned, two other NRPSs directly involved in the elongation of the peptide are PvdJ and PvdD, catalyzing the addition of L-Lys, L-hfOrn, and two L-Thr residues [63,64,65]. The activity of two auxiliary enzymes, PvdG and MbtH, appears to be required for the intra-cytoplasmic maturation of the ferribactin. In detail, MbtH has been proven to enhance the adenylation activity by A domains from many NRPS enzymes [66], while the role of PvdG has not yet been fully elucidated. PvdG could be a trans-acting thioesterase for PvdL and/or PvdI [48]. In addition, in specific strains, different auxiliary enzymes can catalyze tailoring reactions during ferribactin biogenesis. As an example, during the biosynthesis of type II PVD, PvdY_II_ catalyzes the acetylation of N-hydroxy-ornithine, which is supposed to be the preliminary step for the formation of the terminal N-hydroxy-cyclo-ornithine residue, and the subsequent peptide release from the NRPS machinery [55,67]. The acylated ferribactin is then transported to the periplasm through the activity of PvdE, a specific ABC transporter homolog to MacB ABC transporter for cyclic peptides [68]. In the periplasm, PVDI is subjected to diacylation by Ntn-type hydrolase PvdQ [69].

To achieve a functional pyoverdine, two copper-dependent oxidoreductases, PvdP and PvdO, provide for the cyclization and the formation of the final PVDI chromophore by catalyzing the first and last oxidative step, respectively [70,71]. The correct functioning of PvdP requires the activity of two auxiliary enzymes. The activity of PvdP was, in fact, closely associated with the presence of PvdM, which is believed to be indispensable for the incorporation of copper within PvdP [72]. The action of the PvdP seems to be subordinated even to the activity of one periplasmic membrane-associated oxidoreductase, CcmC, which transfers the electrons generated during oxidation to a periplasmatic redox-active compound [48]. However, further mutagenicity studies are required to better elucidate the role played by CcmC.

After the formation of the mature chromophore, the peptide undergoes a rearrangement of the side chains, especially at the level of the α-carboxy and α-amino groups of the L-Glu present in position 1, which can be replaced, depending on the strains investigated, by succinamide, succinate, or α-ketoglutarate, and, less frequently, malamide and malic acid, or even traces of intramolecular cyclized succinic acid. Little is known about why these modifications are made, but it is predicted that they can lead to an advantage according to the different environmental conditions surrounding the microorganism [55,73]. The first enzyme discovered as having an active role in the modification of side chains is PvdN, necessary for the addition of succinamide and succinic acid [73]. Similarly, the PtaA enzyme, periplasmic transaminase A, was associated with the presence of α-ketoglutarate on the PVDI side chain [74]. Although malamide and its derivative, malate, can probably be produced starting from succinamide, the biosynthesis of the two compounds remains currently unknown.

Finally, once the side chain replacements are completed, the mature PVDI is secreted from the periplasm into the external environment through efflux pumps. To date, only two ATP-dependent efflux pumps are known, PvdRT-OpmQ and MdtABC-OpmB. These mediate the passage of PVDI through the membrane [46,75]. However, it is certain that other transporters play a role in the transport of PVDs to the outside environment, because, following the creation of *Pseudomonas* strains with deletions for the two pumps, PVD secretion was not interrupted.

## 3. Biological Functions of PVDs

### 3.1. Iron Uptake

In the bacterial environment, PVDs produced by *P. aeruginosa* PAO1 chelate ferric iron, yield the PVDI-Fe^3+^ complex [49], are recognized at the bacterial surface, and are internalized in the periplasm across the outer membrane by two TonB-dependent transporters, FpvAI and FpvB [76,77,78].

Once in the bacterial periplasm, iron released from PVDI involves numerous proteins encoded by the fpvGHJKCDEF genes and a mechanism of iron reduction. Recently, the overall interaction network between the various proteins encoded by the fpvGHJKCDEF genes has been reported, by using systematic bacterial two-hybrid screening. The authors identified a large protein machinery composed of five interacting proteins, showing (i) an interaction between the two inner-membrane proteins FpvG and FpvH; (ii) an FpvJ–FpvC–FpvF periplasmic complex; and (iii) the interaction of the periplasmic complex (FpvC, FpvF, and FpvJ) with the inner-membrane FpvG–FpvH complex, probably mediated by FpvJ, found to interact with both periplasmic and membrane proteins [46]. In detail, the PVDI-Fe^3+^ complex is first bound by the two periplasmic proteins, FpvC and FpvF [79]. The iron is then released from the PVDI in the bacterial periplasm, not by chemical modification of the siderophore, but by an iron reduction by the FpvG inner membrane reductase [80,81,82]. In the mechanism of iron release, FpvC is proposed as the Fe^2+^ chelator from PVDI, which, subsequently, brings the ferrous iron to the ABC transporter FpvDE for its translocation into the cytoplasm across the inner membrane [79,82,83].

Moreover, after iron release, the apo form of PVDI is most likely bound to the periplasmic siderophore binding protein FpvF, which is able to interact with PvdT, the inner-membrane protein of the PvdRT-OpmQ efflux pump, allowing the recycling of apo-PVDI to the extracellular medium, with the capacity to chelate Fe^3+^ again [46,84,85]. Interestingly, PVDs also mediate the interactions underlying the regulations between microbial communities. Indeed, it is known that PVDs can be used by bacterial PVD non- and low-producers (cheaters) for the supply of iron, as they are capable of encoding receptors compatible with heterologous PVDs produced from other community members [86,87]. Interestingly, environmental drivers, such as iron concentration, organic carbon composition, and pH, can affect PVDs, and thus bacterial interactions [88]. In detail, bacteria reduce PVD production as the iron concentration increases [89,90], particularly when iron is linked to weak organic chelators [91], and when, at low pH, the solubility of iron is increased [92].

### 3.2. PVDs’ Role in Pseudomonas Virulence

In addition to satisfying the iron requirements of the *Pseudomonas* strains, PVDs are also important for the infection and colonization of the host by pathogens towards animals or plants, and therefore, they play a prominent role in the pathogenicity of the opportunistic human pathogen *P. aeruginosa*.

First, due to their extremely high affinity for ferric iron (10^32^ M^−1^) [93], PVDs give the pathogen a significant advantage over the host in their competition for iron, a nutrient essential for growth. In fact, they can remove iron from mammalian iron-sequestering proteins, such as lactoferrin and transferrin [51,94].

Besides iron scavenging, PVDs have several other implications in the virulence of *Pseudomonas* strains. Biofilm formation and PVD production in *P. aeruginosa* have a complex and bidirectional regulatory relationship. PVDs are responsible for obtaining extracellular iron, which is a fundamental nutrient for biofilm formation in different species of bacteria, including *P. aeruginosa* [95]. Notably, Kang and Kirienko demonstrated a fine, bidirectional gene regulatory relationship between PVDs and biofilm, suggesting that biofilm formation is necessary for PVD production. In fact, the biofilm inhibitor 2-amino-5,6-dimethylbenzimidazole attenuated PVD production and rescued C. elegans from *P. aeruginosa*-mediated pathogenesis [96]. Interestingly, transcriptomic analyses indicated that PVDs also seem to be involved in other aspects of the *P. aeruginosa* pathogenesis, including quorum sensing (QS) and the response to reactive oxygen species (ROS) [97,98]. It has been also demonstrated that the *Pseudomonas* quinolone signal (PQS) biosynthetic protein, PqsA, is indispensable for PVD production in a biofilm-dependent manner. This behavior was specific to PqsA and not dependent on the biosynthesis of PQS. However, exogenous PQS promoted a biofilm-independent PVD production [95].

The study of several *P. aeruginosa* mutants, carrying deletions in genes involved in virulence, iron acquisition, or QS, demonstrated a correlation between PVD production of *P. aeruginosa* in airways of immunocompromised patients with cystic fibrosis and its antifungal activity against Aspergillus fumigatus by interfering in biofilm production. This suggests the involvement of PVDs in *P. aeruginosa*’s capability of interfering with fungi, which are competitors in its ecological niches, such as the human microbiome [99].

Moreover, the ability of PVDs to induce biofilm formation plays a central role in the development of antibiotic resistance. *P. aeruginosa* strains especially can resist various antimicrobials as well as the immune system defence mechanism due to the physical/chemical barrier offered by the biofilm [96]. To this end, recent studies [100,101] have been able to highlight a close relation between PVD production and the ability to escape antimicrobial therapies, since the authors found, among 54 clinical isolates showing antibiotic resistance, 93% were PVD producers.

PVDs can also mediate virulence by stimulating the production of exotoxin A and protease PrpL [102], capable, respectively, of inducing cell death by apoptosis and degrading surfactants and interleukins 22 necessary for the immunity of the pulmonary mucosa [103,104,105]. It has also been demonstrated that PVDs are an essential colonization factor in the important plant pathogen *P. syringae*, as they are required to produce tabtoxin [106].

PVDs can also directly exert a toxin-like behavior. A pathogenesis model based on *C. elegans* suggested that the PVDs exert cytotoxicity by removing host iron, inducing mitochondrial damage, mitophagy, and a lethal hypoxic crisis in the host [107,108]. Kang and colleagues [109] demonstrated that PVDs transiently translocase into the host where they binds and, therefore, remove significant amounts of ferric iron. As a consequence of the PVD-mediated iron extraction, host mitochondria, which are iron-rich organelles, suffer damage, compromising their electron transfer and ATP production function, ultimately activating mitochondrial turnover via autophagy.

In addition, in vivo experiments conducted with *P. aeruginosa* showed that the manipulation of the availability of iron-scavenging PVDs affects the bacterial load in infections of the greater wax moth *Galleria mellonella* larvae in complex ways, triggers differential host responses, and alters the expression of regulatory-linked virulence factors [110].

## 4. Inverting the Tide: PVDs as Target for Antimicrobial Therapy

*P. aeruginosa*’s threat as an opportunistic human pathogen is worsened by its great resistance to many common antibiotics. *P. aeruginosa* possesses different resistance mechanisms, including efflux pumps, decreased membrane permeability, enzymes that modify or degrade antimicrobials, and a biofilm that forms a physical barrier; moreover, they can acquire genetically encoded resistance determinants from other pathogens [111]. For these reasons, targeting bacterial virulence factors, minimizing, or preventing, its pathogenicity, is a promising strategy to counteract infection, without necessarily killing the bacteria and reducing the development of newly acquired resistance mechanisms.

As reported in detail in the preceding paragraph, PVDs play a major role in the virulence apparatus of *P. aeruginosa*. Therefore, manipulating PVDs and disrupting their ability to bind iron directly interferes with the principal regulatory mechanisms of *P. aeruginosa*’s virulence, severely compromising its pathogenicity [112].

Different PVD inhibition approaches have been investigated up to now (Table 2).

Various studies have performed high-throughput screenings of compound libraries in search of PVD blockers since PVD-quenching compounds also inhibit their activity. It was observed that anti-PVD compounds can quench the fluorescence of pyoverdines in cell-free filtrates, limit the expression of PVD-dependent genes, and improve *C. elegans* survival after exposure to *P. aeruginosa* strains [112,113].

In a recent study [111], four compounds, namely 3-hydrazinylquinoxaline-2-thiol (LK10), 1,2,3,6,7,8-hexahydro-pyrene-1,3,6,8-tetrone (LK11), 3-amino1,4-dihydroxy-quinoxaline-2-carbonitrile (LK12), and 5E)-5-[(dimethylamino)methylidene]-3-(methyl-sulfanyl)-4,5,6,7-tetrahydro-2-benzothiophen-4-one (LK31), were tested on bacteria cultures and bacteria-free filtrates. LK11 and LK31 acted as anti-virulents while LK10 and LK12 also inhibited *P. aeruginosa* growth. It remains unclear how these small molecules interact with PVDs, but the authors suggest they could bind to the conserved dihydroxyquinoline chromophores.

Functional inhibition of PVDs operated by PQ3 (5-oxo-3-phenyl-4-[2-(1,3-thiazol-2-yl) hydrazin-1-ylidene] pyrazole-1-carbothioamide) was studied by Wang and colleagues [113], identifying it as a PVD blocker. The authors tested a commercially available analogue of PQ3 that was discovered to be an even more effective anti-PVD agent. To characterize its intermolecular interaction, and to provide insights on the molecular recognition mechanism PVD-ligand, they performed nuclear magnetic resonance (NMR), molecular dynamics (MD), and docking simulations. The interaction site resulted in a shallow groove where the compound, the regions of the chromophore, and D-Ser1 and L-Arg2 of the oligopeptide backbone enter in contact, forming a strong electrostatic interaction. Since the compound quenched all three types of PVDs and the chromophore is shared by all fluorescent Pseudomonads, the authors speculate that this binding site could be conserved in pyoverdines from many strains.

A second approach investigated is the functional inhibition of PVDs with iron-mimetic inhibitors. The ability of gallium to disrupt *P. aeruginosa* growth and inhibit biofilm formation is well known [113]. However, in human serum (HS), the pathogen proved less susceptible to gallium-mediated growth inhibition [116]. The reason why this occurs depends on the ability of *P. aeruginosa* to use its proteases and PVD systems to hydrolase HS proteins and acquire iron from the host, counteracting gallium inhibition.

A mimic strategy can also be used by synthetic PVD analogs with a siderophore-like activity. Synthetic analogues should possess the same characteristics of natural PVD, though structurally simpler than native siderophores [124]. Antonietti and collaborators [117] synthetized two analogs close to the endogen PVD of *Pseudomonas* PaO1 with few differences in the peptidic sequence. The analogs had no antibiotic activity and one of them, named aPvd3, showed high iron-chelating properties forming a stable 1:1 complex with iron in water, while its penetration was ruled by a transport membrane protein. According to this evidence, aPvd3 meets the criteria for application as a PVD analogue; it also possesses functional groups that enable the grafting of antibiotic moieties, and could be used for selective delivery in resistant bacteria using a “Trojan Horse strategy”.

Furthermore, inhibition of PVDs could be addressed from a biosynthetic point of view, searching for compounds able to block enzymes of the PVD biosynthetic pathway. A famous class of PVD inhibitors are fluoropyrimidines [111]. Fluoropyrimidines include 5-fluorocytosine (5-FC) and 5-fluorouridine (5-FU) known to have anti-pathogenic and anti-PVD activities, though the exact mechanism of action is unclear [118]. Phenylthiourea (PTU) is another compound able to inhibit the enzyme PvdP tyrosinase, required for the maturation of the PVD chromophore [70], and consequently, to interfere with the production of PVDs [125]. The inhibition mechanism was explained analyzing the crystal structure of apo-PvdP and its interaction with phenylthiourea. Moreover, the presence of a C-terminal lid with a Tyr residue at the end of the active site of apo-PvdP was reported. PTU was able to form hydrogen bonding and hydrophobic interactions with the residues between the N-terminal β-barrel domain (BBD) and the C-terminal tyrosinase domain (TYD) of PvdP, blocking the substrate-binding pocket and interfering with its enzymatic activity. In this way, PTU shows a non-competitive inhibitory activity through the rearrangement of the C-terminal lid.

The above-mentioned anti-virulence approaches are PVDs specific. However, anti-virulence compounds, such as allicin and gallein, have been known to interfere with chronic virulence factors, such as QS and biofilm formation, and produce a broader inhibition effect indirectly influencing the production of pyocyanin, PVD, elastase, rhamnolipids, motility, and toxins [126,127].

A cocktail of different antivirulents, affecting both QS, biofilm formation, and PVDs, together with conventional antimicrobials, could display a synergistic and more effective mechanism able to prevent infections.

## 5. Biotechnological Potential of PVDs

### 5.1. Applications

The metal chelation ability of siderophores can be useful for several medical applications, and PVDs were recently used in diagnostics (Figure 2). The PVD produced by *P. aeruginosa* PAO1 was labelled with Gallium-68 for specific imaging of *Pseudomonas* infections through positron emission tomography (PET). Results displayed specific accumulation in the infected tissues and better distribution than with the clinically used 18F-fluorodeoxyglucose and 68 Ga-citrate [128]. Alternatively, it is possible to employ the PVDs’ iron uptake systems as a device to increase the bacterial uptake of antibiotics. The first attempt was made in 1998, in which two PVDs from *P. aeruginosa* and *P. fluorescens* were linked to ampicillin.

The resulting products were tested against two ampicillin-resistant strains of *P. aeruginosa*, displaying high antimicrobial activity [129]. However, other attempts have been unsuccessful, and the main reasons are represented by the stability of the dual system PVDs-antibiotics, the different compartments in which PVDs and the antibiotics act, and the selectivity of *P. aeruginosa* outer membrane transporters for the different types of PVDs, which hamper the correct uptake of the conjugated antibiotics [130,131,132]. A different type of “Trojan horse” approach is represented by the “cheater invasion”, in which MDR pathogen populations were infiltrated by strains with built-in weaknesses (Figure 2). For example, recently, antibiotic-resistant *P. aeruginosa* strains were replaced by the antibiotic-sensitive *P. aeruginosa* PAO1; this result was achieved by deleting its FpvB PVD receptor, which gives them a selective advantage [133].

PVDs, as virulence factors, also demonstrated to be a promising target to detect the presence of pathogens. A very innovative finger-based sensor integrated on a glove revealed the presence of *P. aeruginosa* by detecting PVDs up to a limit 1.66 µM [134].

The use of PVDs is not limited to human health care but embraces multiple biotechnological applications. PVDs possess several characteristics that make them suitable for agricultural purposes (Figure 2). For example, they can provide plants with nutrients, selectively triggering the mobilization of specific metal ions. In fact, PVDs produced by *P. aeruginosa* can increase the phytoavailability of nickel instead of cadmium in hydroponics [135]. Moreover, they can enhance plant growth and induce plant pathogens suppression. For example, PVD 2112, produced by *P. fluorescence* 2112, can reduce 70% of the phytopathogenic damage made by the fungus *Phytophthora capsici* when directly spread on the plant *Capsicum annuum* [136]. Beyond iron acquisition, PVDs can also regulate gene expression in plants. In fact, the PVD from *P. fluorescence* C7R12 is not only able to restore the growth in Arabidopsis thaliana plantlets in iron deficiency conditions, but it can up-regulate the expression of genes related to the iron acquisition, repressing the defense-related expression of genes [137].

Thanks to their ability to bind metals, PVDs can also be used in bioremediation (Figure 2). For example, they can remove iron from asbestos waste, altering the structure of the asbestos fibers. PVDs from *P. mandelii* and *P. fluorescens* efficiently removed iron from chrysotile-gypsum and amosite-gypsum, respectively, in a concentration-dependent manner [138,139]. PVDs have also been proven to form a strong complex with metals of the actinides series uranium U(VI), curium Cm (III), and neptunium Np(V). Notably, Np(V) forms with the PVD from *P. fluorescence* CCUG 32,456, a complex 1.8 times more stable than EDTA, which is known to be a strong chelator of actinides [140]. PVDs can also desorb cesium (Cs) from illite. In particular, the PVD produced by *P. fluorescens* ATCC 17,400 mobilized Cs from illite as a consequence of the direct ion exchange between carbocation of the PVD chromophore and Cs+9, and weathering illite by iron chelation [141]. In addition, the same PVD was recently utilized to enhance phytoextraction of Cs from illite by the red clover Trifolium pratense, which extracted five times more Cs in the presence of PVD [142].

Siderophores, including PVDs, have been applied in the set-up of biosensors for the detection and monitoring of metal ions, and antibiotics in environmental, clinical, and other samples (Figure 2) [143]. A first successful attempt was made in the early 1990s, when PVD immobilized on pore glass and packed in a flow cell was applied to detect the iron levels in tap and mineral waters. The biosensor showed good stability, but a modest detection limit [144]. The introduction of different immobilization approaches/systems (e.g., sol-gel glass) has shown to increase the biosensor sensitivity detecting low concentrations of iron in different matrices as reported in a recent review [143]. PVDs have been applied to reveal the presence of antibiotics, such as furazolidone. As a matter of fact, this molecule is used to counteract bacterial infections in animals, especially in the aquaculture and poultry field, but it has become illegal due to its genotoxic, carcinogenic, and mutagenic effects; however, it is still used. Furazolidone can quench the fluorescence of PVDs and this effect was successfully exploited by Yin et al. for the efficient and rapid detection of this antibiotic [145,146]. The same authors applied this quenching strategy to successfully detect copper in different samples, such as drinking water, seawater, and biological samples [147].

### 5.2. Cluster Engineering and Heterologous Expression of PVDs

The promising biotechnological potential of PVDs has generated a growing demand of such molecules. Moreover, seeing as each PVD can display specific activities, the generation of new PVDs has been explored. As described above (see paragraph 2), their core structure is synthesized by NRPSs, which are arranged in a modular manner. This organization then allows the possibility to alter the core structures of the peptide by altering the enzymes of a given module with others that have specificity with another amino acid. This strategy was already explored for different peptides with some success and was also applied to PVDs by Winn and colleagues [65] who tried to modify the *P. aeruginosa* PVD. The gene PvdD contains two modules, both incorporating a threonine at the C-terminal. In their attempt they tried to substitute: (1) only the adenylation domain (A); (2) both the A domain and the condensation domain (C). Surprisingly, when only the A domain was substituted, they were able to detect PVD always containing threonine, even if the A domain was specific for other amino acids. Instead, when the A-C substitution was performed, they were able to obtain two new PVDs incorporating lysine or serine in place of the threonine. Despite low levels of production, this success also highlights a more complex role of the condensation domain in NRPSs [148]. In a follow up work [149], the role of the thiolation (T) module was investigated in the synthesis and reported a high portability of this domain. The researchers performed 18 T domain substitutions, and in si cases the T was placed upstream of a C domain and reported PVDs expression at different levels in 15 cases out of 18. This finding has important implication for NRPS engineering, suggesting domain C to be the key point for the generation of recombinant PVDs.

From a biotechnological point of view, higher production levels are mandatory to obtain high levels of products. Native producer strains generally yield poor amounts of products because the synthesis of these secondary metabolites is tightly regulated. For these reasons, heterologous expression represents a valid strategy for the production and the study of new siderophores [150,151], and was also applied for clusters obtained from metagenomes [152].

However, at the time of writing, there are no examples in the literature describing the recombinant expression of PVDs in heterologous hosts. The reason probably lies in the high complexity of PVD synthesis involving different genes among the NRPS core and the other auxiliary genes necessary for the maturation, as shown above. Moreover, due to the product bioactivities, the heterologous hosts need to be highly tolerant to the PVDs. For these reasons, a good solution might be represented by *Pseudomonas putida*. This bacterium is emerging as a promising platform to produce many secondary metabolites, including NRPS [153]. Furthermore, it is a natural producer of PVD [154], and unlike *P. aeruginosa*, is not a human pathogen, allowing cultivation on large scale with few limitations. With the continuous improvement of DNA cloning and editing tools, *P. putida*, in the next few years, could become the elected production platform for natural or synthetic PVDs.

## 6. Analytical Methods for the Identification of PVDs

### 6.1. Colorimetric and Electrochemical Detection

The presence of PVDs within complex matrices can be probed by colorimetric methods, including the non-specific Chrome Azurol S (CAS) assay, widely used for the detection of siderophores independent of their structures (Figure 3A).

The CAS assay was established by Schwyn and Neilands [155]. In this procedure, the CAS and the hexadecyltrimethylammonium bromide (HDTMA) are employed to form the indicator CAS/HDTMA and complex with Iron (III) coming from a FeCl_3_ solution, leading to the production of a blue dye. If a strong iron chelator (e.g., siderophore) binds the iron removing it from the dye, the blue solution turns to orange, thus revealing the presence of a siderophore [156]. Unfortunately, the CAS assay does not allow for the distinguishing of different classes of siderophores. Hence, specific colorimetric tests, such as the Csaky’s and Arnow’s assays, have been developed to detect hydroxamate- and catechol-type siderophores, respectively (Figure 3A) [157]. PVDs react positively in the Csaky’s assay, as expected for the presence of the hydroxamate function. However, the Arnow’s test gives ambiguous results with these molecules, despite bearing a catechol function. While most authors report that PVDs do not react positively to this test [37,73], there is no shortage of evidence that PVDs give positive results with both Csaky’s and Arnow’s assays, indicative of mixed-type siderophores [158].

Furthermore, PVDs can also be detected thanks to their intrinsic fluorescence (λ_Abs_ = 405 nm and λ_Emi_ = 460 nm) as being chromogenic metabolites, due to the presence of the 2,3-diamino-6,7-dihydroxyquinoline chromophore (Figure 3A). Therefore, fluorescence-based assays have been designed using PVDs as effective biomarkers of *Pseudomonas* spp. infections in different tissues, such as burn wounds [159].

Recently, a sensitive and selective electrochemical sensor has been developed for PVD detection in water and body fluids [160]. This sensor is made of a composite material based on exfoliated graphene and gold nanoparticles deposited on a graphite screen-printed electrode and has been employed to determine PVD concentration as function of the current intensity for its electrochemical oxidation.

### 6.2. Mass Spectrometry and Structural Analysis

Numerous technologies have been employed in the structural elucidation of ionophores, including siderophores. Among others, the most employed are certainly UV–visible absorption, NMR, and X-ray diffraction (Figure 3A) [161,162]. However, although these techniques are efficient for the characterization of a single pure molecule, these are low-throughput technologies; thus, being often ineffective on complex mixtures containing hundreds and hundreds of metabolites, and above all, unable to identify metabolites featuring ion chelation properties. Advances in mass spectrometry (MS) technologies, together with the development of chemoinformatic tools, are currently delivering new high-throughput solutions to detect siderophores as well as other ionophores. Among MS-based approaches, the thin-layer chromatography matrix-assisted laser desorption ionization mass spectrometry (TLC-MALDI-MS) has been widely employed for the analysis of bacterial siderophores (Figure 3A). Although it is an old technique, TLC allows for the quick location of iron ligands, as TLC plates can be unveiled with the CAS assay reagent [163]. Inductively coupled plasma mass spectrometry (ICP-MS) coupled with a liquid chromatography (LC) instrumentation represents another valuable resource for the search of ionophores, as this has different advantages as compared to other MS ion sources, including: (i) a higher ion density, which drastically increases its sensitivity (<1 picomole of iron); and (ii) a lower influence of solvents and salts, that, on the other hand, negatively affects electrospray source (ESI) sensitivity [164,165].

Based upon the existence of four stable iron isotopes, i.e., ^54^Fe (5.80%), ^56^Fe (91.72%), ^57^Fe (2.20%), and ^58^Fe (0.28%), several isotopic pattern-based methodologies have been developed for the selective detection of iron ligands. However, an in-depth analysis of untargeted tandem mass data of proton- and metal-bound siderophores reported by Aaron and colleagues [166] revealed that the large majority of siderophores have not been detected in their holo (iron bound) form, but in their apo (protonated) form. In addition, even when the holo form is present, it usually shows a different retention time and MS/MS fragmentation as compared to the apo form, thereby making their linkage a difficult matter [167,168]. To overcome these issues, a native LC-MS based method was recently set up and integrated with the ion identity molecular networking (IIMN), a chemoinformatic tool clustering related molecules from crude extracts based on: (i) MS/MS spectra similarity; and (ii) mass differences enabling the connection of protonated adducts with their relevant metal adducts (e.g., with iron adducts in the case of siderophores) [169]. Basically, after chromatographic separation, this method involves post-column pH adjustment to physiological values and metal infusion, aiming to trigger the formation of metal-bound rather than protonated adducts.

A complementary approach to isotopic pattern-based methods, is screening MS/MS spectra for fragment ions and/or neutral losses diagnostic of siderophore substructures [170]. LC–MS/MS analysis using an all-ion fragmentation (AIF) approach, has been demonstrated to be a valuable strategy for PVD detection in complex matrices by generating characteristic fragment ions, which can be traced back to the corresponding precursor ions based on their retention times. As PVDs feature a typical chromophore (Figure 1), a common strategy is targeting the 2,3-diamino-6,7-dihydroxyquinoline fragment ion at *m*/*z* 204.0773 (C_10_H_10_N_3_O_2_^+^), arising from the retro-Diels–Alder reaction of the chromophore (Figure 3(BI)). However, this strategy cannot be extended to complex extracts containing different PVD isoforms, which may differ for the side chain and/or the chromophore core and, therefore, give MS/MS spectra either showing a low-intensity diagnostic ion at *m*/*z* 204.0773 or even lacking it (Figure 3(BII)). For this reason, this method has been implemented relying on the biological precursor of PVDs, the ferribactin, whose fragmentation typically yields the fragments at *m*/*z* 305.1614, 170.0930, and 136.0762, which can be used as PVD fingerprints together with, or as an alternative to, the 2,3-diamino-6,7-dihydroxyquinoline fragment. In this regard, Rehm and co-workers showed that searching for the ferribactin fragment ion turned out to be useful for the detection of several PVD variants from *Pseudomonas* spp., including unusual congeners bearing an isopyoverdine chromophore. Moreover, these authors developed a mass calculator and fragmentation predictor Excel tool for fast dereplication of PVDs, thus fostering discovery of novel derivatives by HR-MS based approaches [171].

## 7. Conclusions

Thanks to their excellent iron-binding properties, PVDs confer competitive advantages to the producing strains and have different direct and indirect correlation with the virulence of pathogen strains. In this context, besides the canonical iron chelation, an analysis of the literature review allowed us to highlight the preponderant role played by PVD in biofilm formation, one of the main virulence vectors of strains belonging to the genus Pseudomonas. Here, we reported and described new compounds and their related molecular mechanisms capable of reducing the virulence of PVDs, avoiding the use of classic antibiotics. Iron mimicking inhibitors, synthetic PVD analogs, and metabolic disruptors have shown excellent potential in reducing bacterial infections.

Nonetheless, PVDs exhibit promising biotechnological applications on human health and environmental care. Indeed, PVDs have been explored as a “trojan horse” to deliver antimicrobial/anticancer compounds, as biosensors for pathogens, antibiotics, and other molecules, but also for other applications, such as bioremediation and phytostimulation.

However, the utilization of PVDs is somehow limited by their availability and price. At the time of writing, only the pyoverdine from *Pseudomonas fluorescent* is commercialized at a considerably high price (200 euro/mg). This is due mainly to the poor production rate in native strains. A heterologous expression system for PVDs is still lacking, and from a synthetic biology point of view, only few attempts were made to generate new synthetic clusters. The use of a non-pathogenic *Pseudomonas* strain as a microbial cell factory could be the key to unlocking the high-value production of PVDs. Substituting the enzymes that synthesize the peptide core or the “decorating enzymes” using native PVD clusters could be a promising strategy to vary the products, while the deletion of regulating sequences or receptors could help to increase production yield. The discovery of new PVDs with different properties is instead hampered by analytical methods. Classical biochemical assays or MS-based approach are too unspecific, but only few high-throughput analytical methods for the straightforward identification of pyoverdines have been developed until now. These rely upon the targeted search for diagnostic fragments of the highly conserved PVD chromophore, but there is the need to develop a more comprehensive method for the localization of unusual PVD variants. Molecular networking analysis of untargeted high-resolution tandem mass spectrometry data [172] could be a suitable approach for rapid identification of novel PVDs, but a dedicated tandem mass spectral library of already known PVDs is still required. The improvement of rapid and reliable analytical techniques, combined with the generation of microbial cell factories for high production of PVDs, will be the key to finally unlock the biotechnological potential of these valuable molecules.

## Figures and Tables

**Figure 1 ijms-23-11507-f001:**
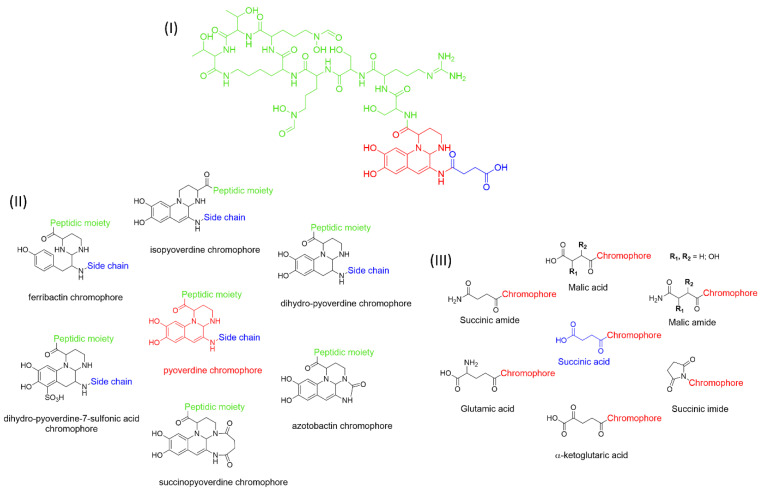
Structure of pyoverdine type 1 produced by *P. aeruginosa* (**I**); examples of different PVDs’ chromophore structures (**II**); illustration of different side chains characteristic of PVDs (**III**).

**Figure 2 ijms-23-11507-f002:**
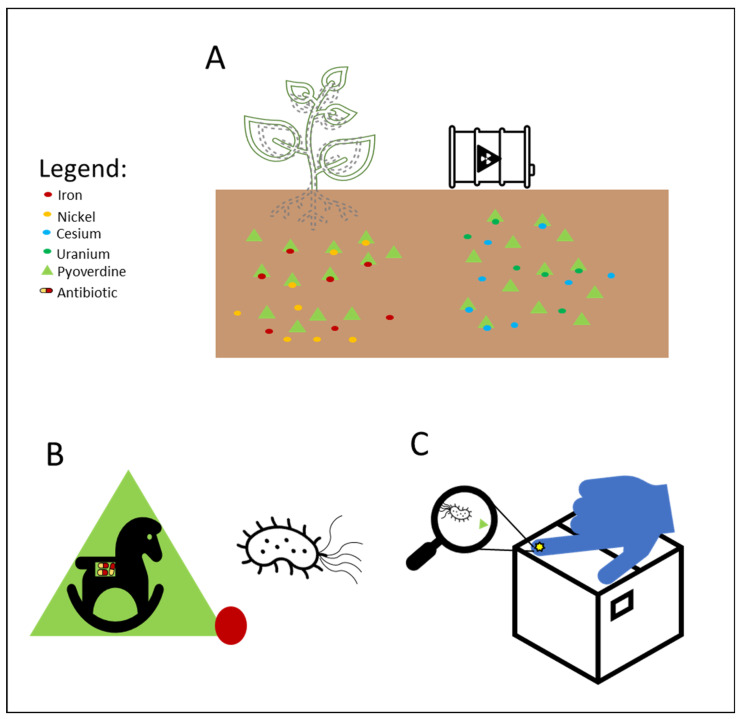
Biotechnological applications of PVDs: (**A**) phytoavailability and bioremediation. PVDs metal binding properties can improve the availability of metal ions necessary for plant growth and help remove toxic metal ions from polluted soils; (**B**) Trojan horse strategy. Iron-complexed PVD bound with an antibiotic can overcome the pathogen drug resistance machinery; (**C**) biosensors. PVDs can be used as a proxy to detect the presence of *P. aeruginosa* on solid surfaces using screen-printed sensing gloves.

**Figure 3 ijms-23-11507-f003:**
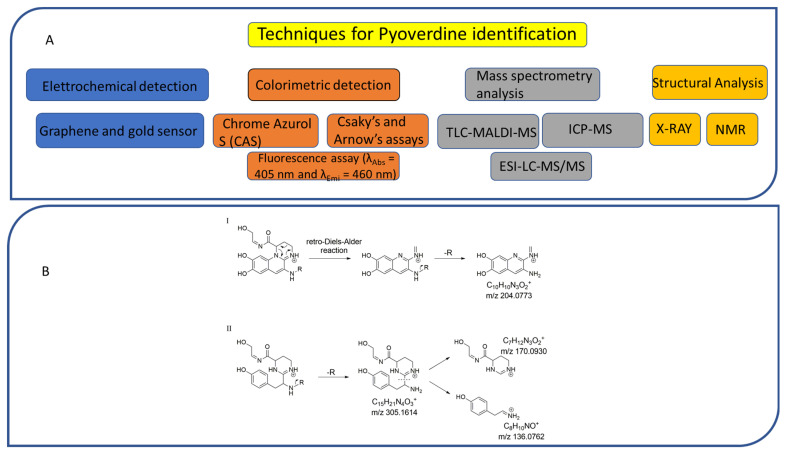
(**A**,**B**). Flow diagram reporting techniques and methods for PVD identification (**A**); MS fragmentation pathways leading to the formation of diagnostic fragment ions from the pyoverdine (**I**) and ferribactin (**II**) chromophores, as reported by Rehm and colleagues (171) (**B**).

**Table 1 ijms-23-11507-t001:** Representative list of siderophores belonging to the class of hydroxamate, catecholate, carboxylate, and mixed ligands.

Type of Siderophore	Structure	Organism	References
**Hydroxamate Siderophores**
Albomycins	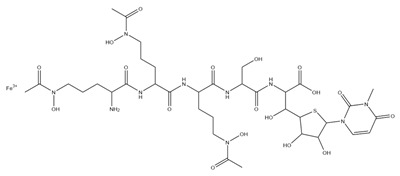	Actinomyces suibtropicus	[8]
Alcaligin	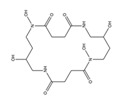	*Bordetella pertussis*; *Bordetella bronchiseptica*	[9]
Bisucaberin	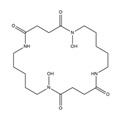	*Alteromonas haloplanktis* SB-112	[10]
Coprogen	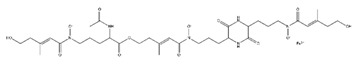	*Trichoderm hypoxylon*	[11]
Ferrichrome	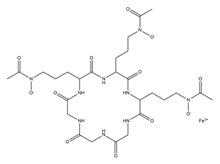	*Lactobacillus casei*	[12]
Ferricrocin	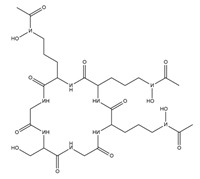	*Trichoderma virens*	[13]
Danoxamine	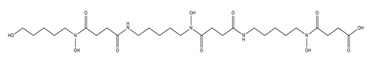	*Streptomyces violaceus* DSM 8286	[14]
Deferoxamine B	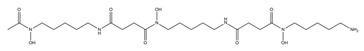	*Streptomyces pilosus*	[15]
Desferrioxamine E(nocardamine)	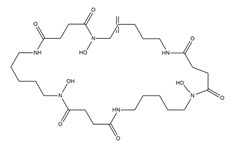	*Streptomyces griseus*	[16]
Fusarinine C	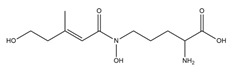	*Fusarium roseum*	[17]
Ornibactin	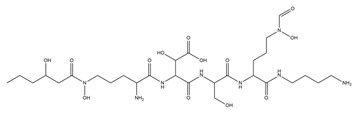	*Burkholderia cepacia*	[18]
Rhodotorulic acid	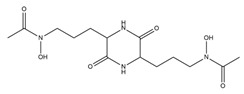	*Rhodotorula pilimanae*	[19]
**Catecholate Siderophores**
Azotochelin	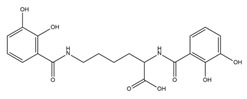	*Azotobacter vinelandi*	[20]
Bacillibactin	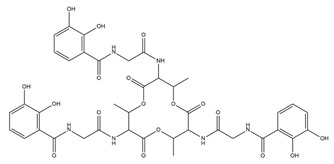	*Bacillus subtilis, Corynebacterium glutamicum*	[21]
Enterobactin	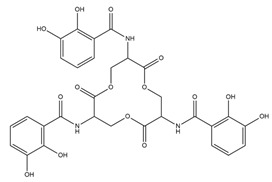	*Escherichia coli*	[22]
Paenibactin	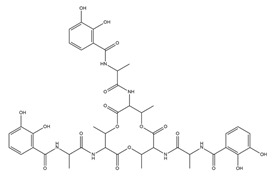	*Paenibacillus elgii* B69	[23]
Protochelin	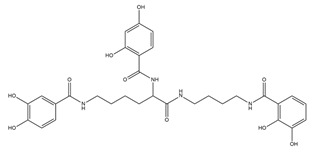	*Azotobacter vinelandi*	[24]
Salmochelin	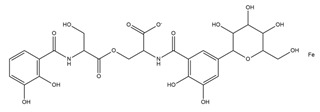	*Salmonella enterica*	[25]
Vibriobactin	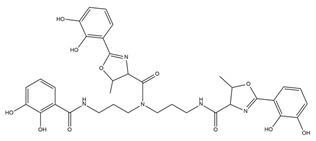	*Vibrio cholerae*	[26]
**Carboxylate Siderophore**
Achromobactin	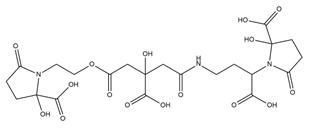	*Erwinia chrysanthemi*	[27]
Rhizobactin	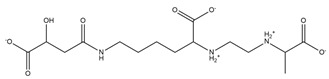	*Rhizobium meliloti*	[28]
Rhizoferrin	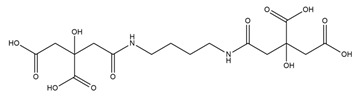	*Rhizopus microsporus*	[29]
Staphyloferrin A	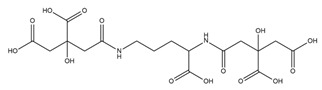	*Staphylococcus hyicus* DSM20459	[30]
**Mixed Ligands**
Aereobactin	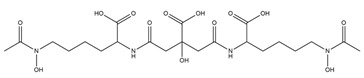	*E. coli*	[31]
Amychelin	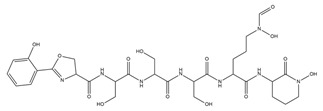	*Amycolatopsis* sp. AA4	[32]
Azotobactin	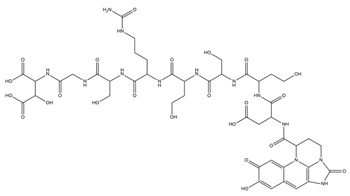	*Azotobacter vinelandii*	[33]
Gobichelin A and B	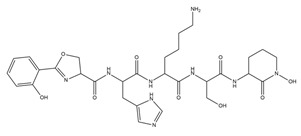	*Streptomyces* sp. NRRL F-4415	[34]
Mycobactins	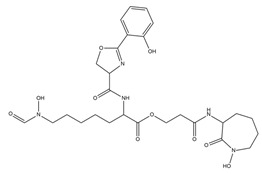	*Mycobacterium tuberculosis*	[35]
Pseudochelin A	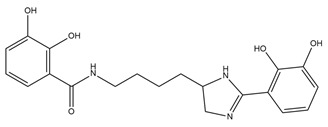	*Pseudoalteromonas piscicida* S2040	[36]
Pyoverdine	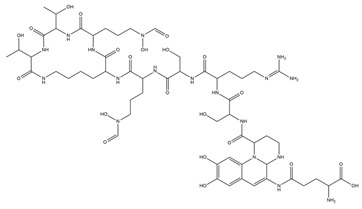	*Pseudomonas aeruginosa*	[37]
Rhodobactin	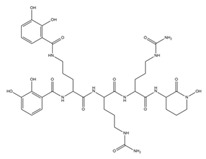	*Rhodococcus rhodochrous* strain OFS	[38]
Yersiniabactin	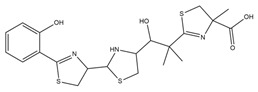	*Yersinia pestis*	[39]

**Table 2 ijms-23-11507-t002:** List of pyoverdine inhibitors and related mechanism.

Pyoverdine Inhibitor	Inhibition Mechanism	Reference
LK10: 3-hydrazinylquinoxaline-2-thiol	Pyoverdine blocker	[111]
LK11: 1,2,3,6,7,8-hexahydro-pyrene-1,3,6,8-tetrone	Pyoverdine blocker	
LK12:3-amino1,4-dihydroxy-quinoxaline-2-carbonitrileN	Pyoverdine blocker	
LK31: (5E)-5-[(dimethylamino)methylidene]-3-(methyl-sulfanyl)-4,5,6,7-tetrahydro-2-benzothiophen-4-one	Pyoverdine blocker	
LK31 analog: (5E)-5-[(dimethylamino)methylidene]-3-(methylsulfanyl)- 4-oxo-4,5,6,7-tetrahydro-2-bemzothyphene-1-carbonitrile	Pyoverdine blocker	
PQ3: 5-oxo-3-phenyl-4-[2-(1,3-thiazol-2-yl) hydrazin-1-ylidene] pyrazole-1-carbothioamide	Pyoverdine blocker	[113]
PQ3 analog: (E)-3-methyl-5-oxo-4-(thiazol-2-yldiazenyl)-2,5-dihydro-1H-pyrazole-1-carbothioamide	Pyoverdine blocker	
Gallium	Iron-mimic	[114,115,116]
aPvd3 analog of PaO1	Pyoverdine-mimic	[117]
5-FC: 5-fluorocytosine	Pyoverdine biosynthesis inhibition	[102,111,118,119]
5-FU: 5-fluorouracil	Pyoverdine biosynthesis inhibition	[111,118,119]
5-FUR: 5-fluorouridine	Pyoverdine biosynthesis inhibition	[119,120]
N2- Succinyl-L-ornithine	Pyoverdine biosynthesis inhibition	[121]
Actinomycin X_2_	Unknown	[122]
Actinomycin D	Unknown	
Eugenyl acetate	Unknown	[123]

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
