# Peer review of "Novel Insights on Pyoverdine: From Biosynthesis to Biotechnological Application"

_ijms, 2022, doi:10.3390/ijms231911507_

Round 1
Reviewer 1 Report
This review highlights PVDs biosynthesis and pathogenic mechanisms. New compounds and their related molecular mechanisms capable to reduce the virulence of PVDs were described.
Chemical nomenclature should be checked, lines 279,280.
Author Response
#Reviewer 1
Comments and Suggestions for Authors
This review highlights PVDs biosynthesis and pathogenic mechanisms. New compounds and their related molecular mechanisms capable to reduce the virulence of PVDs were described.
Chemical nomenclature should be checked, lines 279,280.
Answer to reviewer:
We thank the reviewer for taking the time to carefully read this review. Corrections were made as required.

Reviewer 2 Report
This review described the role of siderophores produced mostly by member of the genus Pseudomonas. The authors decribed biosynthesis, biological functions of PVDs, antimicrobial activity and analytical methods for identyfication of PVDs.
Befor this manuscript will be accepted for publication the authors could make small corrections.
1. In the first chapter I saw the sentence ".....siderophores, small molecules with a mass between 200 and 2000 Da able to chelate iron." As I know the siderphores are known as a ligands for many other metal ions. Please see the publications of Elzbieta Gumienna-Kontecka.
2. The part of analitycal methods can be interesting for many researchers but I think that for clarity it could be broken down into sub-sections.
Author Response
#Reviewer 2
Comments and Suggestions for Authors
This review described the role of siderophores produced mostly by member of the genus Pseudomonas. The authors decribed biosynthesis, biological functions of PVDs, antimicrobial activity and analytical methods for identyfication of PVDs.
Befor this manuscript will be accepted for publication the authors could make small corrections.
1. In the first chapter I saw the sentence ".....siderophores, small molecules with a mass between 200 and 2000 Da able to chelate iron." As I know the siderphores are known as a ligands for many other metal ions. Please see the publications of Elzbieta Gumienna-Kontecka.
2. The part of analitycal methods can be interesting for many researchers but I think that for clarity it could be broken down into sub-sections.
Answers to reviewer:
We thank the reviewer for suggestions that improve the quality of the manuscript.
The changes made according to the reviewer indications are listed below.
- We modify the text as suggested by the reviewer. We added a reference (n°4) and adjusted the final list of references.
- We divided the last section (analytical methods) in two subsections to help the readers to better understand the overall meanings of the section.

Reviewer 3 Report
Dell’Anno et al. prepared a review article highlighting the biosynthesis, pathogenic mechanisms, therapeutic applications, and biotechnological applications of Pyoverdines, as well as modern techniques for the detection of such biomolecules in complexes for drug discovery. The manuscript is easy to follow and the narrative flows well. The English usage is clear and professional. The review is quite comprehensive, so readers can learn all aspects of Pyoverdines. There are also sufficient references provided. I therefore recommend this manuscript to be published in Int. J. Mol. Sci. I only have a few comments on this manuscript:
(1) The Figures can be more nicely-made. The font sizes and figure legends can be larger.
(2) Keyword: more keywords should be provided to make this manuscript more searchable for potential readers. Besides, “biotechnology” is such a vague term, so I would recommend replacing it with keywords that well-represent the topics of this manuscript.
(3) Line 30: It seems like this sentence is missing a verb. I would recommend adding “that are” before the word “able”.
(4) Page 3, bottom three rows: why are there isolated double bonds in those structures?
(5) Figure 2A: if the authors want to show plant growth, then the smaller plant may be shown in dashed lines and the larger (grown) plant may be shown in solid lines.
Author Response
#Reviewer 3
Comments and Suggestions for Authors
Dell’Anno et al. prepared a review article highlighting the biosynthesis, pathogenic mechanisms, therapeutic applications, and biotechnological applications of Pyoverdines, as well as modern techniques for the detection of such biomolecules in complexes for drug discovery. The manuscript is easy to follow and the narrative flows well. The English usage is clear and professional. The review is quite comprehensive, so readers can learn all aspects of Pyoverdines. There are also sufficient references provided. I therefore recommend this manuscript to be published in Int. J. Mol. Sci. I only have a few comments on this manuscript:
(1) The Figures can be more nicely-made. The font sizes and figure legends can be larger.
(2) Keyword: more keywords should be provided to make this manuscript more searchable for potential readers. Besides, “biotechnology” is such a vague term, so I would recommend replacing it with keywords that well-represent the topics of this manuscript.
(3) Line 30: It seems like this sentence is missing a verb. I would recommend adding “that are” before the word “able”.
(4) Page 3, bottom three rows: why are there isolated double bonds in those structures?
(5) Figure 2A: if the authors want to show plant growth, then the smaller plant may be shown in dashed lines and the larger (grown) plant may be shown in solid lines.
Answers for reviewer 3
We thank the reviewer for the suggestions enabling us to improve the manuscript. The changes made according to the requirements of the reviewer are listed below.
- The font sizes have been modified according to the reviewer suggestions.
- New keywords have been provided.
- We modified the sentence adding the words required by the reviewer
- We deleted the double bond. They were a mistake.
- We modified the figure as requested by the reviewer.
